Subject Area:
bioinformatics/genetics/genomics/systems biology/cellular biology

Keywords:
genome-wide association study, fine-mapping, causal variants and genes, single-nucleotide polymorphisms, complex traits, polygenic diseases

Author for correspondence:
I. H. Jonkers
e-mail: i.h.jonkers@umcg.nl

†These authors contributed equally to this study.

# A practical view of fine-mapping and gene prioritization in the post-genome-wide association era

R. V. Broekema†, O. B. Bakker† and I. H. Jonkers

Department of Genetics, University Medical Center Groningen, University of Groningen, Groningen, The Netherlands

RVB, 0000-0003-4946-6701; OBB, 0000-0002-1447-1327; IHJ, 0000-0003-2304-7939

Over the past 15 years, genome-wide association studies (GWASs) have enabled the systematic identification of genetic loci associated with traits and diseases. However, due to resolution issues and methodological limitations, the true causal variants and genes associated with traits remain difficult to identify. In this post-GWAS era, many biological and computational fine-mapping approaches now aim to solve these issues. Here, we review fine-mapping and gene prioritization approaches that, when combined, will improve the understanding of the underlying mechanisms of complex traits and diseases. Fine-mapping of genetic variants has become increasingly sophisticated: initially, variants were simply overlapped with functional elements, but now the impact of variants on regulatory activity and direct variant-gene 3D interactions can be identified. Moreover, gene manipulation by CRISPR/Cas9, the identification of expression quantitative trait loci and the use of co-expression networks have all increased our understanding of the genes and pathways affected by GWAS loci. However, despite this progress, limitations including the lack of cell-type- and disease-specific data and the ever-increasing complexity of polygenic models of traits pose serious challenges. Indeed, the combination of fine-mapping and gene prioritization by statistical, functional and population-based strategies will be necessary to truly understand how GWAS loci contribute to complex traits and diseases.

## 1. Introduction

Most, if not all, phenotypic traits and diseases have a genetic component that influences their development, susceptibility or characteristics. Which genetic regions (loci) are linked to phenotypic traits has largely been determined by genome-wide association studies (GWASs) (figure 1*a*). GWASs compare and associate millions of relatively common genetic variants, usually single-nucleotide polymorphisms (SNPs), between a baseline (healthy) population and one with a trait of interest such as type 1 diabetes [1], coeliac disease [2] or height [3]. The trait-associated genetic loci obtained by GWASs are marked by specific variants referred to as marker or top variants. Each marker-variant signifies a haplotype containing many nearby variants that are in high linkage disequilibrium (LD), indicating that they are most likely to be inherited together [4] (figure 1*b*). Over 4000 GWASs have been published since 2002 [5], yielding almost 150 000 marker variant associations to hundreds of traits [6]. However, despite the method's great initial promise, GWASs have not provided immediate insights into the underlying biological mechanisms of each trait due to two major complicating factors.

Firstly, GWASs cannot distinguish the marker-variant signal from that of the other variants that are in high LD. Over 95% of the variants in high LD ($R^2 > 0.8$) are located outside of genes in the non-coding DNA [7] and can be

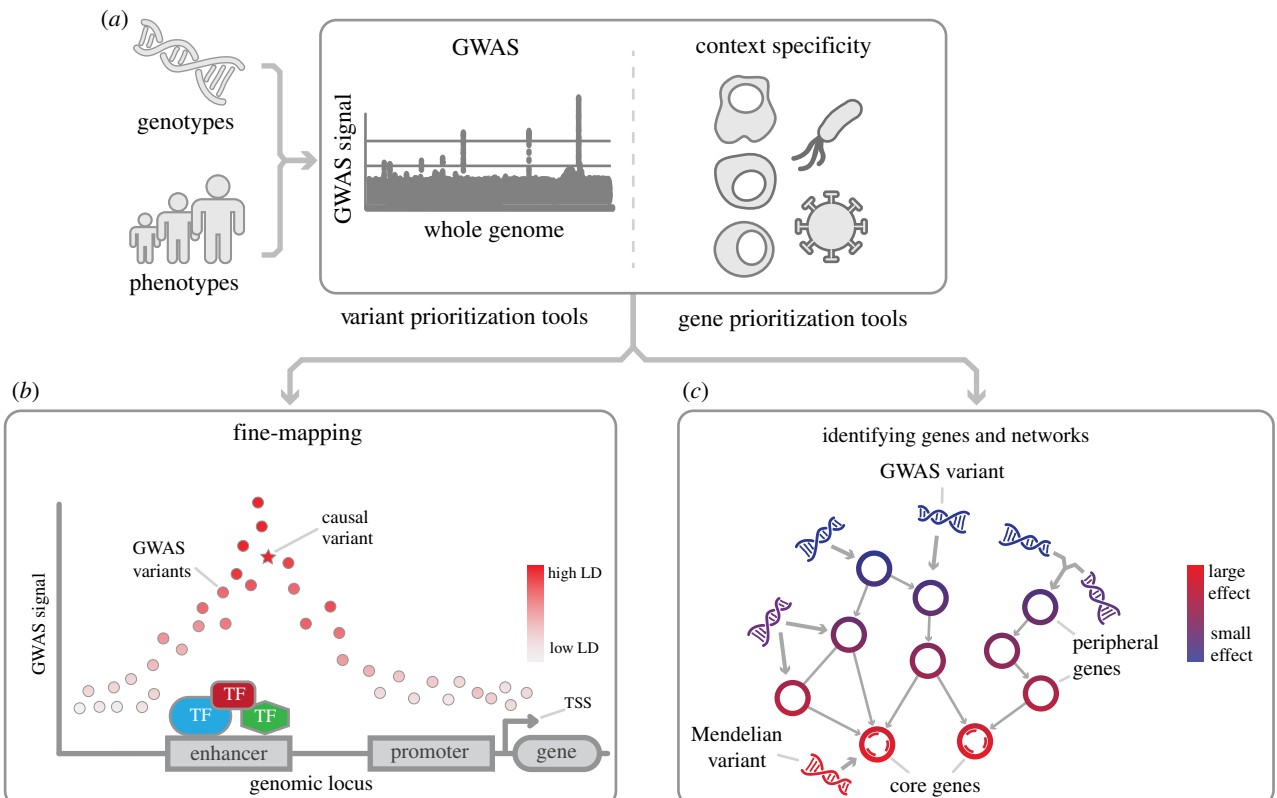

**Figure 1.** Outline of the current post-GWAS workflow. (*a*) First, the correct context needs to be identified for the trait under study. (*b*) Subsequently, causal variants can be fine-mapped to better understand the fundamental mechanisms of transcription. Here, the causal variant (star) is not the strongest GWAS signal, but rather a variant in strong LD with the top effect located in an active enhancer region. (*c*) To gain insights into the biological processes leading to the phenotype, genes can be prioritized and causal networks constructed. GWAS variants are generally common in the population and have smaller effect sizes (blue). Thus, the genes that they impact are more likely to have a small effect on the phenotype as well (peripheral genes). The genes on which many peripheral genes converge (core genes) generally have stronger effects (red) on the phenotype. As such, the variants that affect core genes are more likely to be Mendelian disease variants.

located up to 500 kb apart [8]. Consequently, any of them could be the actual causal variant (figure 1*b*).

Second, the effects of non-coding causal variants can be highly cell-type-, context- and disease-specific [9]. Non-coding DNA contains regulatory regions—enhancers and promoters—that can bind transcription factor (TF) proteins and regulate gene expression [10]. Which enhancers and promoters are used depends on the cell-type-specific abundance of approximately 1600 human TFs and their epigenetically regulated accessibility to a given regulatory region [11]. Variants can disrupt the binding of any of these TFs, resulting in changed enhancer or promoter activity. This, in turn, affects gene expression [12] and cellular pathways [13]. Thus, the cell-type and tissue- or disease-specific micro-environment greatly affect which variants, TFs, genes and pathways are involved (figure 1). These complexities make it difficult to understand how GWAS loci contribute to their associated traits and have significantly hampered the interpretation and application of GWAS results. To address this, many different fine-mapping approaches have been developed in the post-GWAS era with the aim of identifying the important variants and genes and interpreting their biological impact on diseases and traits [14–17].

Important to note is that to reduce fine-mapping complexity, most approaches assume that only a single variant per locus contributes to a trait. This is, however, not a proper reflection of reality as multiple variants within a single GWAS locus can have an effect on a single gene's expression. This can occur in one of two ways: either the effect of the

variants adds up in a linear way (additive effect) or an interaction between two or more variants is required to affect gene expression (epistatic effect) [18,19]. Thus, multiple variants may play a role in a single locus, either within a single cell-type or in a context- and cell-type-specific manner [18]. This further complicates performing and interpreting fine-mapping and gene prioritization approaches. For simplicity, throughout this review, we continue to address variants that affect gene regulation and pathways in association with a GWAS trait in any way as causal, even though a collective of smaller contributing effects acting in unison per locus may be necessary to elicit a functional effect on a GWAS trait.

Here, we assess fine-mapping and gene prioritization approaches that have been used to translate GWAS loci to a functional understanding of the associated trait, while taking cell-type- and disease-specific context into account. Specifically, we review the genetics of lower effect size common variants identified through GWASs rather than high effect-size Mendelian disease variants (figure 1*c*). Moreover, we discuss the impact of the recent paradigm shift towards polygenic models and how these can be used to aid in the identification of gene networks that highlight core disease genes (figure 1*c*).

## 2. Fine-mapping from the variant perspective

Fine-mapping variants in GWAS loci require an understanding of the underlying mechanism by which a variant can

royalsocietypublishing.org/journal/rsob    Open Biol. **10**: 190221

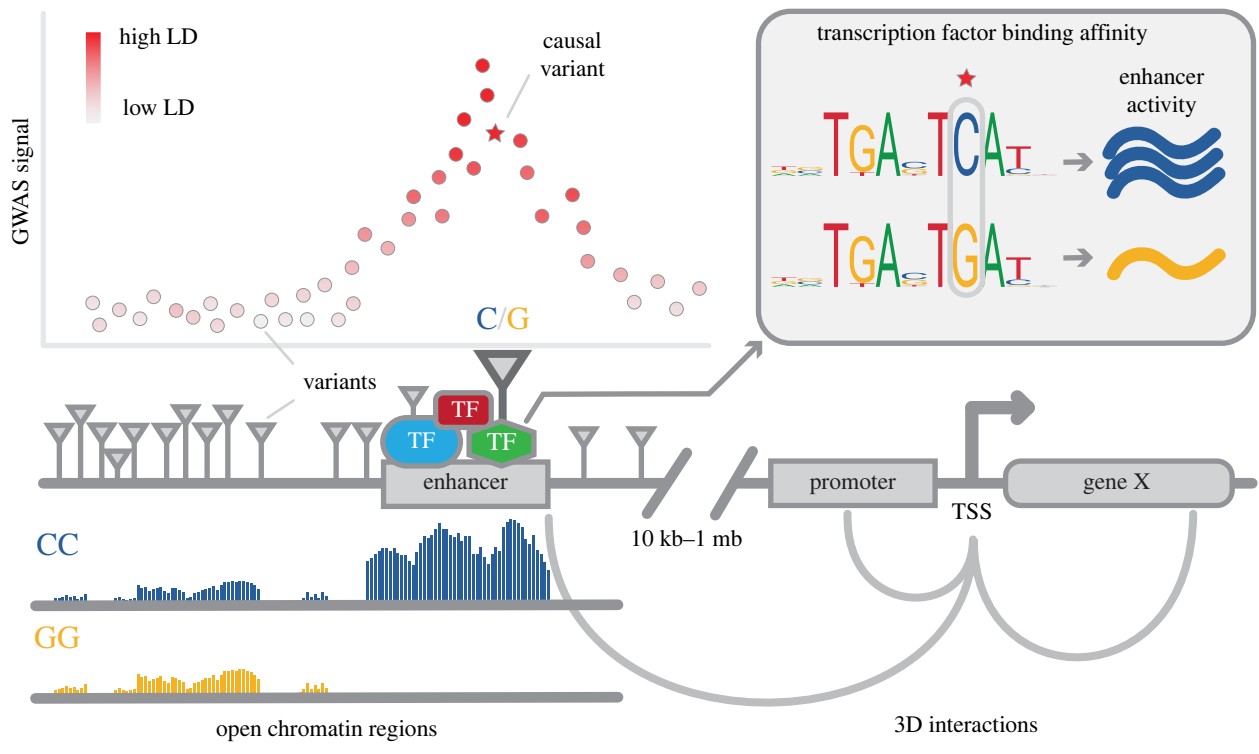

(a) mechanisms by which SNPs can influence enhancer activity

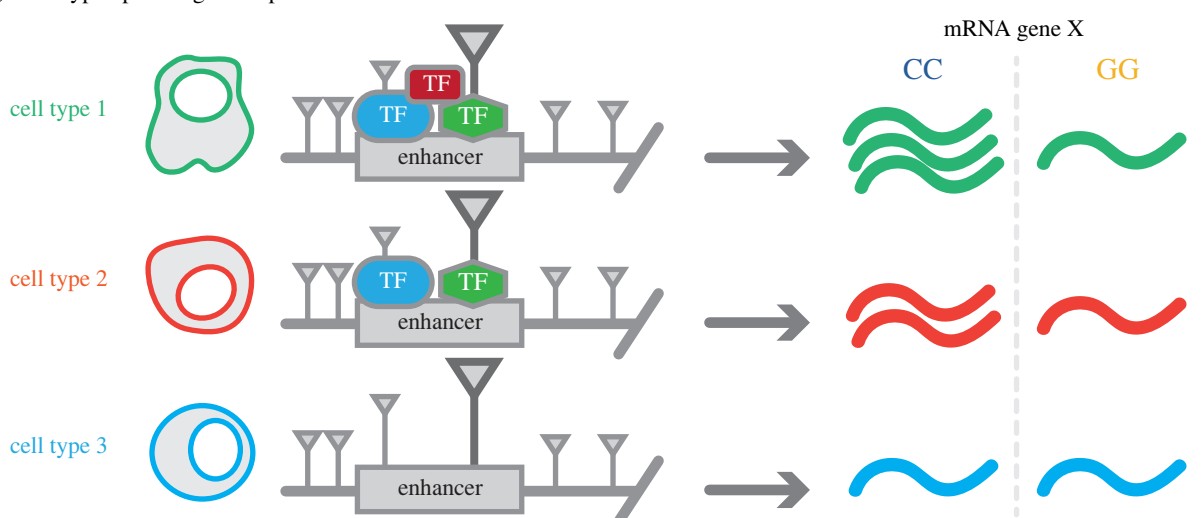

(b) cell-type-specific gene-expression differences

**Figure 2.** An illustrative depiction of a GWAS locus showing example mechanisms by which variant effects on enhancer activity and gene expression can be detected. (a) Many trait-associated variants are shown with varying LD strength (scatterplot) when compared with the GWAS-identified marker variant (in black). In this example, the causal variant is located in an allele-dependent active enhancer (C-allele, caQTL) as shown by the open chromatin regions of the same locus (peak-density plot below the variant). The variant affects the TF binding site of the green TF with a strong binding preference for the C-allele, as shown by the enhancer activity in the 'transcription factor binding affinity' box. In addition, using 3D interactions (grey arches connecting the gene, promoter and enhancer), physical contact with the nearby 'Gene X' indicates the enhancer affects the gene's expression. (b) To highlight cell-type-specific effects, the influence of the causal variant is depicted in three cell types with varying TF availability. The mRNA expression of 'gene X' is stronger for the CC-genotype compared with the GG-genotype because of the increased TF binding affinity to the green TF (as shown in a). This mRNA expression remains low but stable for the GG-genotype in all three cell types regardless of the TF availability but decreases for the CC-genotype in cell types with reduced TF availability, which reduces cooperative TF binding.

contribute to a trait. Overcoming LD and identifying the context-specific variants that are causal to a trait is imperative for understanding disease mechanisms and confidently identifying which downstream genes and pathways are affected. Many functional and computational (high-throughput) fine-mapping methods have been developed and applied for this purpose. Below we review several fine-mapping methods according to their increasing ability to describe the complex role of variants in GWAS traits and diseases.

## 2.1. Identifying overlap with functional elements

The most straightforward fine-mapping approach is to overlap GWAS variants in high LD with functional elements such as promoters and enhancers (figure 2a). Currently, the best resource for functional elements has been compiled by the NIH Roadmap Epigenomics Mapping Consortium [20] (electronic supplementary material, table S1), which used ChIP-seq (electronic supplementary material, table S2)

royalsocietypublishing.org/journal/rsob    Open Biol. 10: 190221

to measure histone marks to determine the location of functional elements in 127 different cell and tissue types [20,21]. Fine-mapping of GWAS variants from 21 autoimmune diseases using the NIH Roadmap and similar data estimated that approximately 60% of candidate causal variants map to immune cell enhancers, and another approximately 8% to promoters [12]. This was also reflected in the tissue-specific enrichment of type 1 diabetes susceptibility variants in lymphoid gene enhancers [22]. Moreover, candidate causal variants were enriched in enhancers defined by the histone mark H3K27ac in specific subsets of CD4+ T cells, CD8+ T cells and B cells [12]. This was also the case in another study in monocytes, neutrophils and CD4+ T cells [23]. Other studies have also identified tissue-specific enrichments of disease-associated variants via overlap with functional elements, showing that this approach can help specify which variants play a role in certain cell types [23,24].

Other ways of detecting regulatory regions that can be used to fine-map GWAS variants are either based on DNA accessibility, such as ATAC-seq [25] and DNase-seq [26] (electronic supplementary material, table S2), or identify the inherent transcriptional activity of enhancers and promoters [27,28], such as GRO-seq [29], PRO-seq [30] and CAGE [31] (electronic supplementary material, table S2). Collective public databases using these techniques—like the NIH Roadmap consortium [20], ENCODE [32], FANTOM5 [33] and the IHEC consortium [34]—are indispensable context-specific resources (electronic supplementary material, table S1). However, it appears to be more difficult than originally anticipated to specify the exact location of regulatory regions since all these methods show different sensitivities and accuracies in the mapping of active regulatory regions [35]. Moreover, overlap of a variant with an active regulatory region may not result in functional disruption of these elements, and thus does not definitively point to causality. This uncertainty limits the accuracy of fine-mapping through overlap with functional elements and still leaves us with a multitude of candidate causal variants.

## 2.2. Inferring allele-specific variant effects

In high-throughput methods such as ATAC-seq, the sequencing reads containing a variant can be separated based on its allele. The allele-specific abundance of sequencing reads can then directly inform us about the functionality of this variant on the open chromatin region. Variants that cause allelic imbalance in regulatory regions are called chromatin accessibility quantitative trait loci (caQTLs; figure 2a) [25,36]. Many caQTLs were identified in primary CD4+ T-cell ATAC-seq peaks, and these showed a strong enrichment in candidate causal autoimmune variants [36]. Similarly, the existence of variants or histone-QTLs that affect regulatory regions by altering enhancer-associated H3K27ac or H3K4me1 histone peaks also implies that these variants have an effect on cell-type-specific enhancer activity [23]. Due to their functional effect on DNA accessibility and epigenetic marks, these variants are more likely to be causal variants for GWAS traits.

Another mechanism by which non-coding GWAS variants can have an allelic effect on gene expression is alternative splicing of genes. GWAS-associated variants have the potential to induce cell-type-specific alternative splicing (sQTL) or could affect trans-acting splicing regulation genes [37,38].

This was shown in a genome-wide approach where 622 exons with intronic sQTLs were identified. One hundred and ten of these exons harboured variants in LD with GWAS marker variants [37]. In a more specific example, the multiple sclerosis-associated PRKCA gene is seemingly affected by an intronic sQTL that increases the expression of a gene isoform more prone to nonsense-mediated decay, thereby reducing the likely protective PRKCA mRNA levels post-transcriptionally [39]. However, sQTLs appear to also act through more complex mechanisms such as indirectly through caQTLs [40], or by inducing alternative upstream transcription start sites [41]. These and many other examples [38] suggest that sQTLs may be an important but complex mechanism by which GWAS-associated variants affect a trait.

## 2.3. Identifying variants that disrupt underlying TF binding sites

Further prioritization of variants in regulatory regions that show allelic imbalances can be done by computational or functional analysis of the underlying TF binding sites (TFBS) or motifs. Regulatory regions consist of both very strict and more degenerate DNA motifs [42] to which TFs can bind in order to initiate local transcription (e.g. enhancer RNAs) and regulate nearby or distant genes [10,27]. Variants can change the TFBS, altering the binding affinity of the TF and changing the activity of a regulatory region (figure 2a) [18,43,44]. The specificity and location of potential TFBSs have been collected for many cell types in large databases such as JASPAR [45], FANTOM5 [33] and ENCODE [32] (electronic supplementary material, table S1), mostly using ChIP-seq and HT-SELEX [46] (electronic supplementary material, table S2).

An enrichment of TFBS disruption by putatively causal variants has been identified for 44 families of TFs [18]. For TFs like AP-1 and the ETS TF-family, regulatory regions containing these disrupted TFBSs also show effects on chromatin accessibility, indicating that the effect of variants on TF binding affinity leads to caQTLs [18]. Similarly, upon identification of nearly 9000 DNase-seq locations affected by allelic imbalances, it was found that the alleles associated with more accessible chromatin were also highly associated with increased TF binding [43]. In a more specific case, TFBS disruption analyses and in vitro confirmation by ChIP-seq led to the identification of rs17293632 as a likely causal SNP that increases Crohn's disease risk by disrupting an AP-1 TFBS [12]. Interestingly, this effect on AP-1 TFBSs was stimulation-specific: H3K27ac peaks with affected AP-1 TFBSs were enriched in stimulated CD4+ T cells compared with non-stimulated cells [12]. This highlights the importance of context-specificity and the need for tissue- and disease-relevant stimulations in experimental set-ups (figure 2b) [12,47]. Finally, in a study of leukaemia patients, a small DNA insertion resulting in a TFBS for MYB created an enhancer near TAL1, which led to activation of this oncogene and the onset of leukaemia [48]. Thus, decreased or increased affinity of TFs due to genetic variants or small DNA changes can have far-reaching effects.

Currently, only 10–20% of the potentially causal non-coding GWAS variants defined by allelic imbalances within a regulatory region can be shown to disrupt a known TFBS [12]. Therefore, the actual causal variants may potentially

act through a different mechanism, or our understanding of TF binding may still be insufficient [49]. One complicating factor here is the potential cooperative binding of more than one TF at an overlapping TFBS. Detection of these cooperative binding motifs is currently being improved by both biological methods (such as SELEX-seq [50]) and computational methods, such as No Read Left Behind (NRLB) [44]) (electronic supplementary material, table S3). A striking example of context-specific cooperative binding of TFs is illustrated by an increased TFBS enrichment of p300, RBPJ and NF-kB in risk loci of GWAS traits as a consequence of the presence of Epstein–Barr virus (EBV) EBNA2 protein [51]. In this study, ChIP-seq data from EBV-transformed B-cell lines were used, together with the RELI algorithm (electronic supplementary material, table S3), to systematically estimate the enrichment of variants in TFBS [51]. In six out of the seven autoimmune disorders tested, RELI identified that 130 out of 1953 candidate causal variants [12] overlapped with EBNA2 binding sites in B-cell lines identified by ChIP-seq [51]. Interestingly, many autoimmune diseases, including coeliac disease and multiple sclerosis [52,53], are thought to be partially triggered by viral infections, suggesting that variants may only be causal when viral factors are also present. Moreover, TF motifs can be highly degenerate, and a small change in TF binding affinity can induce a subtle dosage effect on the activity of a regulatory region [44]. While this effect may be subtle, downstream genes could be affected sufficiently [44] to induce or affect a trait. Thus, a better understanding of how TF binding affinity to DNA motifs is mediated is necessary to comprehend how variants affect the functionality of a regulatory region.

## 2.4. Fine-mapping by detection of regulatory region activity

A more immediate fine-mapping approach is to directly measure the effect a variant can have on the strength of a regulatory region. Active promoters and enhancers have transcription start sites (TSSs), and the activity of an enhancer or promoter is directly correlated with the active transcription from these TSSs [27]. However, some promoter RNAs, and most enhancer RNAs, are very short-lived, making them difficult to detect with most RNA sequencing methods [10,27]. CAGE (electronic supplementary material, table S2) does allow for the identification of exact TSS locations, as well as expression levels of genes, by sequencing 5′-capped transcripts regardless of their stability [30]. CAGE has identified promoter and enhancer effects, and showed that 52% of the effects observed in promoter regions were in secondary CAGE peaks, highlighting that genes can have multiple active promoters depending on the genotype [54]. CAGE QTLs have been observed for loci associated with systemic lupus erythematous (SLE) and inflammatory bowel disorder [54], supporting their relevance in immune disease.

Reporter-plasmid assays can also be applied to directly measure the effects of variants on enhancer or promoter TSS activity by moving variant-containing DNA fragments from their natural environment to a plasmid and transfecting these into a cell type of interest. The most traditional reporter-plasmid assay, the luciferase assay (electronic supplementary material, table S2), was used to confirm a functional effect of rs1421085, which is associated with obesity risk, by showing

that the risk-allele induces an increase in enhancer activity [55]. However, high-throughput reporter assay methods with high resolution are required to fine-map all potentially causal variants within entire GWAS loci based on regulatory region activity.

One such method, the massively parallel reporter assay (MPRA; electronic supplementary material, table S2), can test over 30 000 candidate variants by synthetically creating 180 bp DNA fragments containing both alleles of a variant with a unique barcode and integrating these into GFP-reporter plasmids that are subsequently transfected into different cell lines [56]. An MPRA was used to identify the expression of 12% (3432) of the 30 000 candidate DNA fragments in three cell lines, with 842 showing allelic imbalances caused by SNPs. Indeed, 53 of these SNPs had previously been associated with GWAS traits [56]. Similar high-throughput fine-mapping methods that use patient-derived DNA instead of synthetically generated DNA sequences are STARR-seq [57] and SuRE [58] (electronic supplementary material, table S2). Using a whole-genome approach, the SuRE method managed to screen 5.9 million SNPs in the K562 red blood cell line, identifying over 30 000 SNPs that affect regulatory regions and allowing for in-depth fine-mapping of SNPs for 36 blood-cell-related GWAS traits [59]. Follow-up research on these reporter assays has identified a causal SNP (rs9283753) in ankylosing spondylitis [56] and another (rs4572196) in potentially up to 11 red blood cell traits [59]. Despite the obvious advantages of high-throughput fine-mapping screens, a major drawback is that these methods are usually applied in cancer or EBV-transformed cell lines. These cell lines can be significantly different from trait-specific tissue-derived cell types [60] and have often accumulated many somatic mutations as a consequence of years of culturing [61]. Thus, the wrong variants may be identified as causal because the relevant cell-type and context-specific effects have not been considered [62].

## 2.5. From causal variant to gene using the 3D interactome

When a causal variant has been identified, the gene expression effects of that variant can be directly assessed by mapping the necessary physical interaction of the regulatory region it affects with its target genes (figure 2a) [63,64]. For example, H3K27ac regions containing autoimmune-disease-prioritized variants were linked to the TSS of genes using HiChIP (electronic supplementary material, table S2) and shown to contain cell-type-specific interactions between the TSS of the IL2 gene and rs7664452 in Th17 cells and between rs2300604 and target gene BATF in memory T cells [63]. Interestingly, for 684 autoimmune-disease-associated variants assessed with HiChIP, 2597 gene–variant interactions were identified, indicating that autoimmune disease variants can regulate a multitude of genes. Moreover, only 14% (367) of these gene–variant interactions were with the gene closest to the variant [63]. Another example of a long-range interaction of a causal variant is that of the previously mentioned rs1421085, which is associated with obesity risk and located in an intron of FTO. TFBS disruption analyses have shown that rs1421085 disrupts the ARID5B TF binding motif and affects the activity of an enhancer that regulates IRX3 and IRX5, genes located 1.2 Mb upstream, instead of

the initially expected co-localized *FTO* gene itself [55,65]. Thus, fine-mapping and interaction analysis has identified additional causal genes in this obesity-associated risk locus.

Hi-C (electronic supplementary material, table S2) is another high-throughput method for identifying specific promoter and enhancer gene interactions [19,66–68]. For example, Hi-C was used to prioritize four rheumatoid arthritis genes by overlapping promoter–gene interactions of various primary immune cells with rheumatoid arthritis GWAS variants [19]. Another study analysed Hi-C datasets of 14 primary human tissues and showed that frequently interacting regions (FIREs) are enriched for disease-associated GWAS variants [68]. However, the resolution limitations of Hi-C and other interaction data make it difficult to precisely pin-point the causal variant within a regulatory region [63,64,68]. In addition, cell-type and environmental effects influence regulatory region interactions with genes, as shown by the fact that 38.8% of FIREs were identified in only one tissue or cell type [68]. Thus, multiple strategies as described here and collected in databases such as the EnhancerAtlas2.0 [69] (electronic supplementary material, table S1) should be combined to confidently fine-map causal variants and link them to genes that play a role in GWAS traits.

# 3. Gene prioritization using GWAS traits

Traditional fine-mapping approaches focus on identifying the causal variants that affect a trait of interest. While very important, knowing which variants are causal does not identify the downstream effects of the variant on the trait. One way to gain such insights is by identifying the genes that are affected by each GWAS locus. Moreover, if the causal genes affected by a locus are known, this can reduce the credible set of potentially causal variants. Recent efforts in systems biology have focused on identifying such causal genes and their downstream effects.

## 3.1. Gene prioritization using expression quantitative trait loci

A more comprehensive approach to identifying the genes affected by a GWAS locus is through the use of quantitative trait loci (QTL; figure 3*a*). While caQTLs are often indicative of a causal variant or regulatory region, a specific subset of QTLs called expression QTLs (eQTL) can be used to identify the genes affected by a GWAS locus [70–72]. The simplest way to perform gene prioritization using eQTL analysis is simply to overlap the marker variant of a GWAS locus with the top eQTL variant. An example of this is an SLE risk variant that is also a *cis*-eQTL for the TF *IKF1*. The eQTL on *IKF1* affected the transcription of 10 genes in *trans* that are all regulated by *IKF1* [70], highlighting this gene as a likely candidate causal gene for SLE. Additionally, these types of effects can be context-specific, as was shown for a *cis*-eQTL on *TLR1* after stimulation of peripheral blood mononuclear cells (PBMCs) with *Escherichia coli* [73]. This *cis*-eQTL was also a strong *trans* regulator of the *E. coli*-induced response network, regulating another 105 genes [73], showing that an eQTL can strongly influence the immune response to pathogens.

However, the top eQTL variant might not always be the same as, or in LD with, the top GWAS marker variant due to noise in the eQTL data [74] or to multiple causal effects on a gene or disease in a locus [75]. As a result, many statistical frameworks have been created to give more accurate estimates of overlap or causality between a GWAS locus and a QTL locus, including FUMA [76], COLOC [77] and Mendelian randomization (MR; electronic supplementary material, table S3). The latter is commonly used to estimate causality between GWAS and QTL profiles [78–84] and has been successfully applied to identify genes causally linked with complex traits [3,79–81]. For example, MR studies were able to identify a causal role for *SORT1* on cholesterol levels [79,81], a role which has been experimentally validated [85]. Still, MR can be challenging as multiple variants in LD can affect the same gene (linkage), and several genes can be affected by the same causal variants (pleiotropy) [70,73,86]. More recent work on MR has focused on more accurately controlling for pleiotropy and linkage [79,81,82,84]. Independent variant selection for MR is currently done by either LD-based clumping or some form of stepwise regression using tools like GCTA's COJO [75] (electronic supplementary material, table S3), which only select for independence and not causality. Accurate fine-mapping can potentially help these efforts by improving the independent variant selection for MR since fine-mapping can reveal the true causal variants independent of linkage.

Recently, it has been suggested that approximately 70% of the heritability in mRNA expression is due to *trans*-eQTLs [87,88], which highlights the importance of *trans*-eQTL relationships. While *trans*-eQTLs have the potential to further our understanding of complex traits, the multiple testing burden is very large due to the large number of comparisons that have to be made when doing genome-wide *trans*-eQTL mapping (in the worst case, millions of variants times approx. 60 000 genes) [70,72]. Therefore, many eQTL studies opt to only map *cis*-eQTL effects genome-wide, as this dramatically reduces the number of comparisons that have to be made [70–72,74]. Another approach is to limit the number of comparisons by only mapping *trans* effects for a predefined subset of variants or genes [70,72,73,86]. However, since a full *trans*-eQTL mapping dataset is rarely available, overlap between *trans*-acting genes and GWAS loci will be missed.

An additional challenge with QTL-based gene prioritization approaches lies in the context-specificity of the QTL data used, as different tissues, cell types, time points and stimulation conditions can induce many different expression patterns and different interactions with the variants in a GWAS locus [23,73,89–92]. Consequently, the QTL information that is available might not be informative for the trait under study. This is especially challenging when studying traits that are present in a tissue other than blood, as is the case for neurological disorders [93,94], because sufficiently powerful cell-type- or context-specific QTL studies are usually not available. However, with the advent of single-cell RNA sequencing (scRNAseq) and the increasing availability of large-scale datasets for tissues other than blood, some of these challenges are being overcome [70,72,90,91]. scRNAseq (electronic supplementary material, table S2) allows for high-throughput eQTL analysis in individual cell types instead of a bulk population, as shown for PBMCs [90]. This allows for an increase in resolution and can help to assess only the trait-relevant cell types [91], as shown for eQTLs on

royalsocietypublishing.org/journal/rsob    Open Biol. **10**: 190221

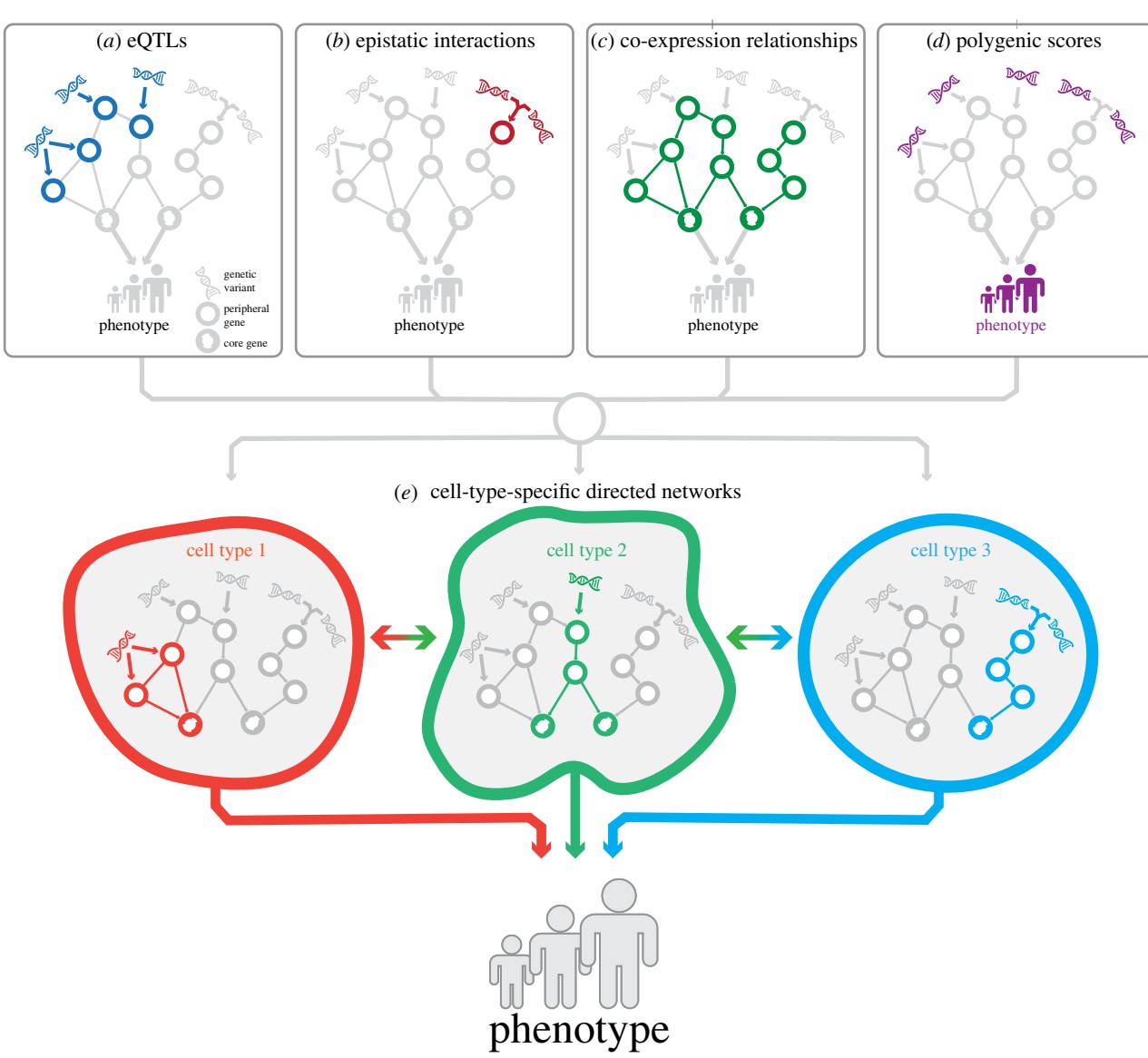

**Figure 3.** Aspects of fine-mapping genes from GWAS loci. (*a*) Using eQTLs (dark blue) and CRISPRi/a-based assays, GWAS loci can be linked to genes when using the correct context. (*b*) Not every relationship between genetics and expression can be described additively. Epistatic effects (dark red) describe a relationship where two (or more) mutations are needed to arrive at the phenotype. (*c*) Using co-expression, regulatory relationships between genes can be quantified, but the specific role of genetics in these relationships is unknown. (*d*) Using PGSs, the joint effects of GWAS loci can be assessed, sacrificing resolution to obtain higher-level insights into the pathways affected by the genetics associated with a phenotype. (*e*) When assessed at single-cell resolution, the total network can be deconstructed into the cell-type relevant components. Affected cells can subsequently display an altered interaction with other cells within a tissue or individual, leading to a changed tissue- or individual-wide outcome for a phenotype.

*TSPAN13* and *ZNF414*, which were only present in CD4+ T cells and not in bulk or other specifically assessed cell types [90]. Consortia that are amassing single-cell data at a large scale in many different tissues—like the Human Cell Atlas [95], Single-cell eQTLgen [96] and the LifeTime consortium [97] (electronic supplementary material, table S1)—will facilitate the use of single-cell sequencing data for traits where bulk RNA-seq obtained from blood is not informative.

## 3.2. Identifying downstream effects of GWAS loci using other QTLs

Beyond gene-expression-based eQTL, a plethora of other QTL types exist that affect the abundance of proteins (pQTL) [98,99], metabolites (mQTL) [100], DNA methylation (meQTL) [101], microbiota (miQTL) [102] and cells (cell-count or ccQTL) [103,104]. Naturally, these can all be overlapped with GWAS loci to obtain insights into their pathology. For example, the *ex vivo* cytokine response to stimulation has been shown to have strong genetic regulators [99]. Interestingly, all the associated effects found were *trans* (i.e. not in proximity to the cytokine genes), suggesting that the release of cytokines is controlled by genes in the receptor's pathways rather than being directly controlled by the mRNA levels of the cytokine. Moreover, context-specificity is important, as QTLs affecting cytokines from T cells were found to be enriched in autoimmune GWAS loci, whereas QTLs affecting cytokines from monocytes were more enriched in infectious-disease-associated loci [99]. Thus, the effects of genetics on traits should not only be studied at the level of gene expression, but also at levels more directly related to a phenotype.

royalsocietypublishing.org/journal/rsob    Open Biol. **10**: 190221

## 3.3. Functional approaches to mapping genetic effects on expression

While eQTL analysis provides invaluable insights into the genes that affect a trait or disease, context- and cell-type-specific biases in the expression data and LD structure in GWAS loci cause potential errors in gene prioritization. With the recent introduction of CRISPR/Cas9-based screens [105] (electronic supplementary material, table S2), it is now possible to functionally validate eQTL effects in a high-throughput manner independent of LD structure and in a cell-type relevant to the trait of interest.

CRISPR-based assays use guide RNAs to bind specific regions of the genome and either activate (CRISPRa) or interfere (CRISPRi) with the transcription of genes or enhancers [106]. Recent advances in both scRNAseq and CRISPRi/a have facilitated methodologies that evaluate enhancer effects on genes in single cells [107]. For example, a recent effort evaluated the effects of 5920 candidate enhancers on gene expression using CRISPRi [107]. Strikingly, 664 showed a significant effect on gene expression in K562 cells. Thus, CRISPRi-based assays are capable of identifying enhancer–gene pairs in a high-throughput manner. However, as only approximately 10% of candidate enhancers were actually found to affect gene expression, identifying which enhancers are active based on already available data might not always be straightforward, even for a very well-characterized cell line such as K562 [20,32,34,58,59].

In addition to mapping active enhancer gene pairs, CRISPRi/a-based assays can be used to identify epistatic interactions between genes and to generate gene networks based on changes in co-expression in perturbed versus non-perturbed cells (figure 3b). Genes that are strongly co-expressed are likely to be regulated by a shared mechanism [86]. Therefore, identifying such genes can help reveal the gene network that leads to a disease-associated trait [94,108,109]. Indeed, a CRISPRi screen that targeted 12 TFs, chromatin modifying factors and non-coding RNAs was able to identify epistatic effects in cells perturbed by two guide RNAs [110]. In these cells, chromatin accessibility remained relatively stable in loci associated with autoimmune disease in cells with one perturbed TF. However, significant changes were observed when evaluating the chromatin accessibility for the same loci in cells also perturbed for NFKB1. This again highlights the importance of taking the entire context of a trait into account when fine-mapping or interpreting the role of a GWAS locus.

A major drawback of the majority of CRISPRi/a screens is that they are very laborious and therefore usually performed in easily manipulated, but also highly modified, cancer cell lines [61]. Fortunately, recent studies have shown that CRISPRi screens can be applied to primary T cells [111,112]. This, while challenging, needs to be extended to other tissues and model systems. These studies will greatly assist variant, regulatory region and gene fine-mapping efforts because they directly identify the active enhancer–gene pairs and the downstream gene network affected in specific cell types. In addition, future work could focus on performing CRISPRi/a screens in patient-derived cells that contain relevant risk genotypes to fully reach variant-level resolution.

## 3.4. Mapping gene–gene regulatory interactions using population data

Co-expression can also be modelled based on inter-individual variation in expression, which can be used to prioritize disease genes and make inferences about the downstream consequences of diseases (figure 3c) [94,108,109,113]. For example, DEPICT (electronic supplementary material, table S3) integrates gene co-regulation with GWAS data to provide likely causal genes and pathways relevant for the trait [113]. Moreover, the GADO tool (electronic supplementary material, table S3) correctly identified causal genes in 41% of a cohort of 83 patients with varying Mendelian disorders, and prioritized several novel causal candidate genes by combining trait-specific gene sets with a co-expression network [109]. Finally, eMAGMA (electronic supplementary material, table S3) used co-expression together with tissue-specific eQTLs in brain regions to prioritize 99 candidate causal genes for major depressive disorder [94]. These co-expression modules were enriched in brain regions but not in whole-blood, highlighting the tissue-specific nature of the co-expression networks [94].

Population-based co-expression networks describe the relationships between genes through both genetics and environment. Consequently, based on the co-expression alone, it is not possible to separate which part of the co-expression is due to genetics. Therefore, these networks have limited use for fine-mapping causal variants and are mainly used to identify genes and pathways affected by GWAS loci after gene prioritizations have been made. In addition, co-expression networks are not directed [108]. Genetic information of the individuals used to generate the co-expression network would solve this issue, as the genetic and environmental components could be separated and directionality could be added into the network [108], although this is not a trivial task. Fine-mapping would be of great value in modelling the genetic component of the network by facilitating the selection of true causal variants.

## 3.5. Fine-mapping under the omnigenic model

As discussed throughout this review, it is becoming increasingly clear that complex traits are highly polygenic and that many variants can deregulate *cis-* and *trans*-acting factors in a variety of ways (figure 2a). In the light of this, Boyle *et al.* [87] proposed an omnigenic model for complex traits in which each gene that is expressed in the cell will have an effect on the trait or disease in some way (figure 1c) [87,88]. For example, height is so polygenic that most 100 kb genomic windows seem to contribute to explaining its variance. Given that the effect sizes of the individual variant are getting so small, it raises the question: what does the causality of the individual variant mean in a complex trait [87,88,114]? If the omnigenic model is true, it presents a major challenge for fine-mapping GWAS loci, particularly for the interpretation of the downstream consequences as the complexity of genetic effects on traits will only increase. In addition, current functional assays may not be suited to model the small and subtle variant effects and gene–gene or gene–environment interactions observed in population studies using millions of individuals.

Instead, the complete GWAS signal from all loci associated with a trait can be used to estimate a polygenic

score (PGS) that describes an individual's genetic predisposition for the given trait. In its most basic form, a PGS constitutes the linear combination of all independent risk genotypes weighted by the GWAS effect size, but many more sophisticated methods exist (figure 3*d*) [115–117]. The PGS for a trait can be associated with the expression level of genes (and proteins) in a population [72,118]. If there are strong correlations, GWAS loci together, as represented by the PGS, are jointly influencing these genes. These genes probably represent core genes in a disease-associated co-expression network. Although PGSs have issues when it comes to broad applicability across populations [119], they can be a useful abstraction layer to make sense of a polygenic trait.

Given we are becoming aware of the likely polygenic and even omnigenic nature of traits, fine-mapping the individual GWAS locus seems like an impossible task. However, with current approaches the stronger, and arguably more important, genetic effects associated with traits and diseases can be elucidated [70,72,73]. Moreover, by using abstraction layers such as PGS, inferences can be made about the joint consequences of these effects [72]. Indeed, the genes and pathways associated with stronger or joint genetic effects are more likely candidates for drug interventions [120] (electronic supplementary material, table S1). Although we might never fully comprehend all the tiny effects and interactions underlying a trait, we will probably see an increase in clever ways to arrive at the interpretable biological mechanisms behind traits.

## 4. Future perspectives

We have reviewed recent high-throughput GWAS fine-mapping approaches that can identify variants and genes causal for a trait or disease. The complexity and uncertainty present in aspects of these approaches illustrates that a single approach does not suffice to grasp the full cause and effect of candidate variants and genes. In addition, while large datasets, mostly in blood, have identified many potentially causal variants and genes associated with traits, these candidates need to be refined and validated using tissue- and cell-type-specific resources in combination with trait-specific environmental factors to recapitulate the true biological state of each trait as closely as possible. An additional challenge lies in translating these disease genes into clinical practice, as prioritized genes might not be existing, nor practical, drug targets.

Despite these challenges, we believe that combining the use of patient-derived material, with methods that find regulatory regions and their downstream genes will aid drug target identification for complex diseases. In addition, this knowledge could be used to generate prediction models that aid in the fast and non-invasive identification of trait-specific variants and genes in the general population. This will form the foundation of our understanding of complex traits, aid drug development and will allow tailored precision medicine in the near future.

Data accessibility. This article does not contain any additional data.
Authors' contributions. R.V.B. and O.B.B. conceived and wrote the manuscript. I.H.J. wrote and critically edited it.
Competing interests. We declare we have no competing interests.
Funding. O.B.B. is supported by an NWO VIDI grant (no. 016.171.047) and an NWO VENI grant (no. NWO 863.13.011). I.H.J. and R.V.B. are supported by a Rosalind Franklin Fellowship from the University of Groningen and an NWO VIDI grant (no. 016.171.047).
Acknowledgements. We acknowledge Kate McIntyre for editorial assistance and critically reading the manuscript.

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
