## [Reviewer comments · Open Biology]

Review History

RSOB-19-0221.R0 (Original submission)

Review form: Reviewer 1

Recommendation

Accept with minor revision (please list in comments)

Do you have any ethical concerns with this paper?

No

Comments to the Author

The paper "A practical view of fine-mapping in the post-GWAS era" from Broekema, Bakker, and Jonkers, is a comprehensive review of the type of data and analyses that are currently being generated to understand the genetic causes of common diseases.

This review is very much needed in the fast moving, and divergent fields of genetical genomics. The review is thorough and up to date. I outlined below the parts that were unclear to me and could benefit from a more thoughtful explanation. The Figures would also benefit from a more clear exposition.

General comments:

1. I may suggest to change the word "fine-mapping" in the title and text, otherwise please clearly define fine-mapping in the context of this review.

Also, specify that you are fine-mapping common snps in population studies, since this review does not include methods for rare variants.

Maybe a more appropriate focus of the review (instead of fine-mapping) could be Advances in genetical genomics to identify causal genes and networks (using population based genomics).

Also, there is an assumption (Line 114-121) of one single causal variant, which may not even be important to note if the focus of the review is modified.

(Instead of "fine-mapping", I would specify that the genetics of common casual variants is used as a tool to refine the genetically-based networks and genes).

2. It would help to add a Table with data input, output, type, analysis and uses (for the genetical genomics/fine-mapping context of this review).

Add to discussion:

3. Add druggable genome: in drug development, one has to also account for genes that are potentially druggable (even once the drug target gene is found to activate the key driver gene), to be able to translate this finding.

4. Mention splicing!

Specific comments:

Line 35-37: rephrase. GWAS associate single SNPs, not haplotypes. Explain the SNP -> region (because of LD) in haplotype.

Line 59-62: rephrase.

Line 71:72: "more successful methods according to their increasing ability to describe the complex role of SNPs in GWAS traits and diseases."

Please define your metrics of success better, is it most successfully proved by validation in experiments in wet lab? in clinical studies?

Line 114-121: This comment is relevant to the whole paper, so it should be placed in intro (and explained better) or in discussion.

This is a huge limitation of all the "fine-mapping" methods.

Line 197-198: "methods that use patient-derived DNA sequences": unclear wording: how does this define only these methods?

Figures comments:

Figure 1c. misleading. eQTLs refer to SNPs. Maybe eGene more appropriate?

Clarify what eQTL is used for - is it for SNP or gene prioritization?

Figure 2: Clarify information. A lot of concepts introduced that are not discussed in the paper: need better discussion to introduce these in text and in the figure:

- what is an enhancer RNA?

- what is the "associated gene" (line 800)?

is the link between enhancer and gene only using 3D-interactions?

how else can we link these two? Possibly mention enhancer-promoter databases such as Nucleic Acids Res. 2019 Nov 19. pii: gkz980. doi: 10.1093/nar/gkz980.

- Figure 2b: is this information from allele specific expression QTL? (specify in the same way in the legend as was done for Figure 2a caQTL)
- Are the TFs not present at all in different cell types? Is this an example where the cooperative binding motifs can be used (ref line 152).

Figure 3. Aspects of fine-mapping genes from GWAS loci.

- Unclear the aim of this figure: is it to summarize the type of information and results achieved by the different datasets?
- Is the "E" panel a subtype incorporating panel "A" and "B"? I would remove and note this in legend.
- Why is the "Fine-mapping" only including the A and B panels?

The authors could try to organize this figure on the bases of a table (see general comment above): to specify data input/analyses (e.g. eQTL/CRISPR), and the type of information that can be extracted from it (Genetic of expression).

Decision letter (RSOB-19-0221.R0)

25-Nov-2019

Dear Mr Bakker

We are pleased to inform you that your manuscript RSOB-19-0221 entitled "A practical view of fine-mapping in the post-GWAS era" has been accepted by the Editor for publication in Open Biology. The reviewer has recommended publication, but also suggest some minor revisions to your manuscript. Therefore, we invite you to respond to the comments and revise your manuscript.

Please submit the revised version of your manuscript within 7 days. If you do not think you will be able to meet this date please let us know immediately and we can extend this deadline for you.

- 1) A text file of the manuscript (doc, txt, rtf or tex), including the references, tables (including

captions) and figure captions. Please remove any tracked changes from the text before submission. PDF files are not an accepted format for the "Main Document".

2) A separate electronic file of each figure (tiff, EPS or print-quality PDF preferred). The format should be produced directly from original creation package, or original software format. Please note that PowerPoint files are not accepted.

3) Electronic supplementary material: this should be contained in a separate file from the main text and meet our ESM criteria (see <http://royalsocietypublishing.org/instructions-authors#question5>). All supplementary materials accompanying an accepted article will be treated as in their final form. They will be published alongside the paper on the journal website and posted on the online figshare repository. Files on figshare will be made available approximately one week before the accompanying article so that the supplementary material can be attributed a unique DOI.

Online supplementary material will also carry the title and description provided during submission, so please ensure these are accurate and informative. Note that the Royal Society will not edit or typeset supplementary material and it will be hosted as provided. Please ensure that the supplementary material includes the paper details (authors, title, journal name, article DOI). Your article DOI will be 10.1098/rsob.2016[last 4 digits of e.g. 10.1098/rsob.20160049].

4) A media summary: a short non-technical summary (up to 100 words) of the key findings/importance of your manuscript. Please try to write in simple English, avoid jargon, explain the importance of the topic, outline the main implications and describe why this topic is newsworthy.

Images

Data-Sharing

It is a condition of publication that data supporting your paper are made available. Data should be made available either in the electronic supplementary material or through an appropriate repository. Details of how to access data should be included in your paper. Please see <http://royalsocietypublishing.org/site/authors/policy.xhtml#question6> for more details.

Data accessibility section

Sincerely,

The Open Biology Team

<mailto:openbiology@royalsociety.org>

Referee:

Comments to the Author(s)

The paper "A practical view of fine-mapping in the post-GWAS era" from Broekema, Bakker, and Jonkers, is a comprehensive review of the type of data and analyses that are currently being generated to understand the genetic causes of common diseases.

This review is very much needed in the fast moving, and divergent fields of genetical genomics. The review is thorough and up to date. I outlined below the parts that were unclear to me and could benefit from a more thoughtful explanation. The Figures would also benefit from a more clear exposition.

General comments:

1. I may suggest to change the word "fine-mapping" in the title and text, otherwise please clearly define fine-mapping in the context of this review.

Also, specify that you are fine-mapping common snps in population studies, since this review does not include methods for rare variants.

Maybe a more appropriate focus of the review (instead of fine-mapping) could be Advances in genetical genomics to identify causal genes and networks (using population based genomics). Also, there is an assumption (Line 114-121) of one single causal variant, which may not even be important to note if the focus of the review is modified.

(Instead of "fine-mapping", I would specify that the genetics of common casual variants is used as a tool to refine the genetically-based networks and genes).

2. It would help to add a Table with data input, output, type, analysis and uses (for the genetical genomics/fine-mapping context of this review).

Add to discussion:

3. Add druggable genome: in drug development, one has to also account for genes that are potentially druggable (even once the drug target gene is found to activate the key driver gene), to be able to translate this finding.

4. Mention splicing!

Specific comments:

Line 35-37: rephrase. GWAS associate single SNPs, not haplotypes. Explain the SNP -> region (because of LD) in haplotype.

Line 59-62: rephrase.

Line 71:72: "more successful methods according to their increasing ability to describe the complex role of SNPs in GWAS traits and diseases."

Please define your metrics of success better, is it most successfully proved by validation in experiments in wet lab? in clinical studies?

Line 114-121: This comment is relevant to the whole paper, so it should be placed in intro (and explained better) or in discussion.

This is a huge limitation of all the "fine-mapping" methods.

Line 197-198: "methods that use patient-derived DNA sequences": unclear wording: how does this define only these methods?

Figures comments:

Figure 1c. misleading. eQTLs refer to SNPs. Maybe eGene more appropriate?

Clarify what eQTL is used for - is it for SNP or gene prioritization?

Figure 2: Clarify information. A lot of concepts introduced that are not discussed in the paper: need better discussion to introduce these in text and in the figure:

- what is an enhancer RNA?

- what is the "associated gene" (line 800)?

is the link between enhancer and gene only using 3D-interactions?

how else can we link these two? Possibly mention enhancer-promoter databases such as Nucleic Acids Res. 2019 Nov 19. pii: gkz980. doi: 10.1093/nar/gkz980.

- Figure 2b: is this information from allele specific expression QTL?

(specify in the same way in the legend as was done for Figure 2a caQTL)

- Are the TFs not present at all in different cell types? Is this an example where the cooperative binding motifs can be used (ref line 152).

Figure 3. Aspects of fine-mapping genes from GWAS loci.

- Unclear the aim of this figure: is it to summarize the type of information and results achieved by the different datasets?

- Is the "E" panel a subtype incorporating panel "A" and "B"?

I would remove and note this in legend.

- Why is the "Fine-mapping" only including the A and B panels?

The authors could try to organize this figure on the bases of a table (see general comment above): to specify data input/analyses (e.g. eQTL/CRISPR), and the type of information that can be extracted from it (Genetic of expression).

Author's Response to Decision Letter for (RSOB-190221.R0)

See Appendix A.

Decision letter (RSOB-19-0221.R1)

05-Dec-2019

Dear Mr Bakker

We are pleased to inform you that your manuscript entitled "A practical view of fine-mapping and gene prioritization in the post-GWAS era" has been accepted by the Editor for publication in Open Biology.

You can expect to receive a proof of your article from our Production office in due course, please

check your spam filter if you do not receive it within the next 10 working days. Please let us know if you are likely to be away from e-mail contact during this time.

Sincerely,

The Open Biology Team
mailto: openbiology@royalsociety.org

Appendix A

Comments to the Author(s)

The paper "A practical view of fine-mapping in the post-GWAS era" from Broekema, Bakker, and Jonkers, is a comprehensive review of the type of data and analyses that are currently being generated to understand the genetic causes of common diseases.

This review is very much needed in the fast moving, and divergent fields of genetical genomics. The review is thorough and up to date. I outlined below the parts that were unclear to me and could benefit from a more thoughtful explanation. The Figures would also benefit from a more clear exposition.

We thank the reviewer for critically looking at the manuscript and are happy that the reviewer recognizes the need and relevance of the review. We have addressed the reviewers comments below in a point by point basis.

General comments

1. I may suggest to change the word "fine-mapping" in the title and text, otherwise please clearly define fine-mapping in the context of this review. Also, specify that you are fine-mapping common snps in population studies, since this review does not include methods for rare variants.

Maybe a more appropriate focus of the review (instead of fine-mapping) could be Advances in genetical genomics to identify causal genes and networks (using population based genomics). Also, there is an assumption (Line 114-121) of one single causal variant, which may not even be important to note if the focus of the review is modified. (Instead of "fine-mapping", I would specify that the genetics of common casual variants is used as a tool to refine the genetically-based networks and genes).

Answer:

We agree with the reviewer that the definition of fine-mapping is somewhat unclear in the original manuscript. In light of this we have made several changes to better reflect the differences between fine-mapping of GWAS SNPs and the downstream interpretation of the genes affected by a GWAS locus. Please find a summary of the changes below:

We have opted to change the title of the manuscript to :

“A practical view of fine-mapping and gene prioritization in the post-GWAS era”

We also agree that the manuscript focuses on common variation, and does not discuss the impact of rare variants in depth. We have added several sentences highlighting that we are focussing on lower effect size common variants rather than high effect size mendelian disease variants. We have also updated figure one to reflect these changes.

In addition, we have changed the final paragraph of the introduction and the abstract to better reflect the link between fine-mapping and gene prioritization efforts and to highlight the complexity of genetic signals

Updated text #1:

Old: line 59-62, page 3):

Here we assess the most successful fine-mapping and interpretation approaches that have been used to translate GWAS loci to a functional understanding of the associated trait, while taking cell-type- and disease-specific effects into account. Moreover, we also discuss the impact on fine-mapping of the recent paradigm shift towards polygenic models.

(New: line 71-77, page 4):

Here we assess fine-mapping and gene prioritization approaches that have been used to translate GWAS loci to a functional understanding of the associated trait, while taking cell-type- and disease-specific context into account. Specifically, we review the genetics of lower effect-size common variants identified through GWASs rather than high effect-

size mendelian disease variants (figure 1C). Moreover, we discuss the impact of the recent paradigm shift towards polygenic models and how these can be used to aid in the identification of gene networks that highlight core disease genes (figure 1C).

Updated text #2:

(Old: line 15-17, page 2):

Here we review several successful fine-mapping approaches that translate GWAS loci into understanding of the underlying mechanisms of complex traits.

(New: line 16-18, page 2):

Here we review fine-mapping and gene prioritization approaches that, when combined, will improve understanding of the underlying mechanisms of complex traits and diseases.

Updated text #3:

(Old: line 23-25, page 2):

Indeed, the combined application of statistical, functional and population-based strategies will be necessary to truly understand how GWAS loci contribute to complex traits and diseases.

(New: line 25-27, page 2):

Indeed, the combination of fine-mapping and gene prioritization by statistical, functional and population-based strategies will be necessary to truly understand how GWAS loci contribute to complex traits and diseases.

The remark regarding the 'single causal variant' assumption in line 114-121 will be answered further along in this document.

2. It would help to add a Table with data input, output, type, analysis and uses (for the genetical genomics/fine-mapping context of this review).

Answer:

We are not entirely certain what the reviewer would like us to add to the current tables that we have generated in the supplementary data for this review. We have 3 supplementary tables that describe all databases currently used for fine-mapping and gene prioritization methods (Suppl. Table 1), a table that specifies many methods that are used to detect regulatory elements on which variants may have an effect and methods that directly measure these effects (Suppl Table 2) and finally a table that compiles the majority of computational and statistical tools available to fine-map and prioritize variants and genes (Suppl Table 3). If the editorial board, disagrees or would consider it more appropriate to move this table from the Supplements to the main text we are happy to do so.

Add to discussion:

3. Add druggable genome: in drug development, one has to also account for genes that are potentially druggable (even once the drug target gene is found to activate the key driver gene), to be able to translate this finding.

Answer:

Indeed, we did not discuss the ‘druggability’ of genes in the original manuscript in depth. We agree with the reviewer that the ability to actually target core genes with drugs, remains a challenge for translating fundamental research to the clinic. As such we have added a sentence to the discussion on drug targeting of genes (see below). We did not discuss this in depth as we feel this is beyond the scope for the review, which focuses more on identifying molecular mechanisms than on translating these findings to the clinic.

Updated text:

(old: line 411-415, page 14):

In addition, while large datasets, mostly in blood, have identified many potentially causal SNPs, genes and co-expression networks associated with traits, these candidates need to be refined and validated using tissue- and cell-type-specific resources in combination with trait-specific environmental factors to recapitulate the true biological state of each trait as closely as possible.

(new: line 434-445, page 15):

In addition, while large datasets, mostly in blood, have identified many potentially causal variants and genes associated with traits, these candidates need to be refined and validated using tissue- and cell-type-specific resources in combination with trait-specific environmental factors to recapitulate the true biological state of each trait as closely as possible. An additional challenge lies in translating these disease genes into clinical practice, as prioritized genes might not be existing, nor practical, drug targets. Despite these challenges, we believe that combining the use of patient-derived material, with methods that find regulatory regions and their downstream genes will aid drug target identification for complex diseases. In addition, this knowledge could be used to generate prediction models that aid in the fast and non-invasive identification of trait-specific variants and genes in the general population. This will form the foundation of our understanding of complex traits, aid drug development and will allow tailored precision medicine in the near future.

4. Mention splicing!

Answer:

We have added several additional sentences discussing the impact of splicing on fine-mapping and the identification of causal genes.

Extra text at the end of chapter 2.2 (new paragraph after line 113 in the old text):

(New: line 128-139, page 5-6):

Another mechanism by which non-coding GWAS variants can have an allelic effect on gene expression is alternative splicing of genes. GWAS-associated variants have the potential to induce cell-type specific alternative splicing (sQTL) or could affect trans-acting splicing regulation genes [37,38]. This was shown in a genome-wide approach where 622 exons with intronic sQTLs were identified. 110 of these exons harbored variants in linkage disequilibrium with GWAS marker-variants [37]. In a more specific example, the multiple sclerosis associated PRKCA gene is seemingly affected by an intronic sQTL that increases the expression of a gene isoform more prone to nonsense-mediated decay, thereby reducing the likely protective PRKCA mRNA levels post-transcriptionally [39]. However, sQTLs appear to also act through more complex mechanisms such as indirectly through caQTLs [40], or by inducing alternative upstream transcription start sites [41]. These, and many other examples [38] suggest that sQTLs may be an important but complex mechanism by which GWAS-associated variants affect a trait.

Specific comments:

Line 35-37: rephrase. GWAS associate single SNPs, not haplotypes. Explain the SNP -> region (because of LD) in haplotype.

Answer:

We agree that GWAS associates single SNPs and believe that the original phrasing in the text captures the relation between SNPs, haplotypes and LD accurately. We feel that we do not state that GWAS associates haplotypes, rather that the SNPs used for GWAS are tagging a haplotype. If the editorial board feels differently we are happy to rephrase this paragraph

Line 33-39 of the manuscript:

GWASs compare and associate millions of relatively common genetic variants, usually single nucleotide polymorphisms (SNPs), between a baseline (healthy) population and one with a trait of interest such as type 1 diabetes [1], celiac disease [2] or height [3]. The trait-associated genetic loci obtained by GWASs are marked by specific variants referred to as marker- or top-variants. Each marker-variant signifies a haplotype containing many nearby variants that are in high linkage disequilibrium (LD), indicating that they are most likely inherited together [4] (figure 1B).

Line 59-62: rephrase.

Answer:

We have re-phrased the paragraph between line 59-62. See the response to general comment #1 for the updated version.

Line 71:72: "more successful methods according to their increasing ability to describe the complex role of SNPs in GWAS traits and diseases."

Please define your metrics of success better, is it most successfully proved by validation in experiments in wet lab? in clinical studies?

Answer:

We agree with the reviewer that the wording of success is not appropriate in this context. Our aim was to indicate that we made a selection out of the large volume of methods and approaches that are out there as we cannot cover all of them. We have changed line 71-72 to the following:

Updated text:

(Old: line 70-72, page 4):

Below we review some of the more successful methods according to their increasing ability to describe the complex role of SNPs in GWAS traits and diseases.

(New: line 84-86, page 4):

Below we review several fine-mapping methods according to their increasing ability to describe the complex role of variants in GWAS traits and diseases.

Line 114-121: This comment is relevant to the whole paper, so it should be placed in intro (and explained better) or in discussion. This is a huge limitation of all the "fine-mapping" methods.

Answer:

We have added a statement about the complexity of genetic signals and the limitation of fine mapping methods into the introduction of the manuscript at line x-x. See also the response to comment #1. Updated 'single causal variant' assumption in line 114-121 → Moved to the introduction, right before the newly updated text for lines 59-62, see comment #1:

Updated text:

(Old: line 114-121, removed):

To reduce fine-mapping complexity, it is easiest to assume that only a single SNP per locus contributes to a trait. However, with over 60,000 common SNPs identified to affect allele-specific DNA accessibility in 114 different cell-types, multiple caQTLs were found within a single GWAS locus [35]. Thus, multiple SNPs may play a role in a single locus, either within a single cell-type, or in a context- and cell-type-specific manner [35]. For simplicity, we continue to address SNPs that affect gene regulation and pathways in association to a GWAS trait in any way as causal, even though a collective of smaller contributing SNP effects acting in unison per locus may actually be necessary to elicit a functional effect on a GWAS trait.

(New: line 60-70, page 3-4):

Important to note, is that to reduce fine-mapping complexity, most approaches assume that only a single variant per locus contributes to a trait. This is however not a proper reflection of reality as multiple variants within a single GWAS locus can have an effect on a single gene's expression. This can occur in one of two ways, either the effect of the variants adds up in a linear way (additive effect) or an interaction between two or more variants is required to affect gene expression (epistatic effect) [18,19]. Thus, multiple variants may play a role in a single locus, either within a single cell-type, or in a context- and cell-type-specific manner [18]. This further complicates performing and interpreting fine mapping and gene prioritization approaches. For simplicity, throughout this review we continue to address variants that affect gene regulation and pathways in association to a GWAS trait in any way as causal, even though a collective of smaller contributing effects acting in unison per locus may be necessary to elicit a functional effect on a GWAS trait.

Line 197-198: "methods that use patient-derived DNA sequences": unclear wording: how does this define only these methods?

Answer:

By "patient-derived DNA sequences" we meant to highlight the difference with methods such as MPRA which use synthetically generated DNA. To better highlight this distinction we changed line 197-198 to the following.

Updated text:

(Old: line 197-199):

Similar high-throughput fine-mapping methods that use patient-derived DNA sequences are STARR-seq [51] and SuRE [52] (electronic supplementary material, table S2).

(New line 215-217, page 8):

Similar high-throughput fine-mapping methods that use patient-derived DNA instead of synthetically generated DNA sequences are STARR-seq [51] and SuRE [52].

Figure comments:

Figure 1c. misleading. eQTLs refer to SNPs. Maybe eGene more appropriate?
Clarify what eQTL is used for - is it for SNP or gene prioritization?

Answer:

We do not fully agree with the reviewer that eQTL's refer solely to SNPs, in our view an eQTL represents the effect of the SNP on the gene, rather than just the SNP as an entity. The eQTL in our view then represents the link between GWAS SNPs and genes. However we realize that this might not be the most common view of an eQTL, thus we have opted to simplify the figure by removing the annotation of the eQTL and adding clearer explanations of these concepts in the figure legends. In addition, we have edited panel C to reflect the new text on rare and common variants, as well as some minor stylistic changes. We hope that with these changes we better introduce the concepts of GWAS variants, mendelian variants, core and peripheral genes.

Figure legend:

(Old, line 791-801, page 27)

Figure 1. Outline of the current post-GWAS workflow. A) Firstly, the correct context needs to be identified for the trait under study. B) Subsequently, causal SNPs can be fine-mapped to better understand the fundamental mechanisms of transcription. C) To gain insights into the biological processes leading to the phenotype, genes can be prioritized and causal networks constructed.

(New, line 821-830, page 28)

Figure 1. Outline of the current post-GWAS workflow. A) Firstly, the correct context needs to be identified for the trait under study. B) Subsequently, causal variants can be fine-mapped to better understand the fundamental mechanisms of transcription. Here the causal variant (star) is not the strongest GWAS signal, but rather a variant in strong LD with the top effect located in an active enhancer region C) To gain insights into the biological processes leading to the phenotype, genes can be prioritized and causal networks constructed. GWAS variants are generally common in the population and have smaller effect-sizes (blue). Thus the genes that they impact are more likely to have a small effect on the phenotype as well (peripheral genes). The genes on which many peripheral genes converge (core genes) generally have stronger effects (red) on the phenotype. As such the variants that affect core genes are more likely to be mendelian disease variants.

Figure 2: Clarify information.

A lot of concepts introduced that are not discussed in the paper: need better discussion to introduce these in text and in the figure:

- what is an enhancer RNA?
- what is the "associated gene" (line 800)?

Is the link between enhancer and gene only using 3D-interactions? How else can we link these two? Possibly mention enhancer-promoter databases such as Nucleic Acids Res. 2019 Nov 19. pii: gkz980. doi: 10.1093/nar/gkz980.

- Figure 2b: is this information from allele specific expression QTL? (specify in the same way in the legend as was done for Figure 2a caQTL)
- Are the TFs not present at all in different cell types? Is this an example where the cooperative binding motifs can be used (ref line 152).

Answer:

We have made several changes to figure 2 to clarify the reviewers questions. In addition, we have updated the figure legend

Figure legend:

(Old, line 791-801, page 27)

Figure 2. An illustrative depiction of a GWAS locus showing ways in which SNP effects on enhancer activity and gene expression can be detected. A) Many trait-associated SNPs are shown with varying linkage-disequilibrium (LD) strength as compared to the GWAS-identified marker-SNP. In this example the causal SNP is located in an allele-dependent active enhancer (C-allele, caQTL) as shown by the open chromatin regions of the same locus. The SNP introduces a transcription factor (TF) binding site for the orange TF with a strong binding preference for the C-allele, as shown by the transcribed enhancer RNA. In addition, using 3D-interactions, physical contact with a nearby gene indicates the enhancer affects the gene's expression. B) To highlight cell-type-specific effects, the influence of the causal SNP is depicted in three cell-types with varying available TFs. mRNA expression of the associated gene remains stable in all cell-types regardless of the available TFs for the GG-genotype but increases with more available TFs for the CC-genotype.

(New, line 832-846, page 28)

Figure 2. An illustrative depiction of a GWAS locus showing example mechanisms by which variant effects on enhancer activity and gene expression can be detected. A) Many trait-associated variants are shown with varying linkage-disequilibrium strength (scatterplot) as compared to the GWAS-identified marker-variant (in black). In this example the causal variant is located in an allele-dependent active enhancer (C-allele, caQTL) as shown by the open chromatin regions of the same locus (peak-density plot

below the variant). The variant affects the transcription factor (TF) binding site of the green TF with a strong binding preference for the C-allele, as shown by the enhancer activity in the 'Transcription factor binding affinity' box. In addition, using 3D-interactions (gray arches connecting the gene, promoter, and enhancer), physical contact with the nearby 'Gene X' indicates the enhancer affects the gene's expression. B) To highlight cell-type-specific effects, the influence of the causal variant is depicted in three cell-types with varying TF availability. The mRNA expression of 'Gene X' is stronger for the CC-genotype compared to the GG-genotype because of the increased TF-binding affinity to the green TF (as shown in part A of this figure). This mRNA expression remains low but stable for the GG-genotype in all three cell-types regardless of the TF availability but decreases for the CC-genotype in cell-types with reduced TF availability, which reduces cooperative TF-binding.

Figure 3. Aspects of fine-mapping genes from GWAS loci.

- Unclear the aim of this figure: is it to summarize the type of information and results achieved by the different datasets?

- Is the "E" panel a subtype incorporating panel "A" and "B"?

I would remove and note this in legend.

- Why is the "Fine-mapping" only including the A and B panels?

The authors could try to organize this figure on the bases of a table (see general comment above): to specify data input/analyses (e.g. eQTL/CRISPR), and the type of information that can be extracted from it (Genetic of expression).

Answer:

We have re-designed figure 3 to make our overarching point more clear. In addition, we have updated the figure legend to reflect these changes

Figure legend:

(Old, line 804-812, page 28)

Figure 3. Aspects of fine-mapping genes from GWAS loci. A) Using eQTL and CRISPRi/a-based assays, GWAS loci can be linked to genes when using the correct context. B) Not every relationship between genetics and expression can be described additively. Epistatic effects describe a relationship where two (or more) mutations are needed to arrive at the phenotype. C) Using co-expression, regulatory relationships between genes can be quantified, but the specific role of genetics in these relationships is unknown. D) Using polygenic scores, the joint effects of GWAS loci can be assessed, sacrificing resolution to obtain higher-level insights into the pathways affected by the genetics associated to a trait. E) Combined with the appropriate context, these aspects can be used to describe the entirety of a causal cascade underlying a trait.

(New, line 848-858, page 28-29)

Figure 3. Aspects of fine-mapping genes from GWAS loci. A) Using eQTLs (dark blue) and CRISPRi/a-based assays, GWAS loci can be linked to genes when using the correct context. B) Not every relationship between genetics and expression can be described additively. Epistatic effects (dark red) describe a relationship where two (or more) mutations are needed to arrive at the phenotype. C) Using co-expression, regulatory relationships between genes can be quantified, but the specific role of genetics in these relationships is unknown. D) Using polygenic scores, the joint effects of GWAS loci can be assessed, sacrificing resolution to obtain higher-level insights into the pathways affected by the genetics associated with a phenotype. E) When assessed at single cell resolution, the total network can be deconstructed into the cell-type relevant components. Affected cells can subsequently display an altered interaction with other cells within a tissue or individual, leading to a changed tissue- or individual wide outcome for a phenotype.